# A systematic analysis of genetic interactions and their underlying biology in childhood cancer

Josephine T. Daub [1,5], Saman Amini[1,5], Denise J. E. Kersjes[1], Xiaotu Ma [2], Natalie Jäger[3], Jinghui Zhang [2], Stefan M. Pfister[3,4], Frank C. P. Holstege [1] & Patrick Kemmeren [1✉]

Childhood cancer is a major cause of child death in developed countries. Genetic interactions between mutated genes play an important role in cancer development. They can be detected by searching for pairs of mutated genes that co-occur more (or less) often than expected. Co-occurrence suggests a cooperative role in cancer development, while mutual exclusivity points to synthetic lethality, a phenomenon of interest in cancer treatment research. Little is known about genetic interactions in childhood cancer. We apply a statistical pipeline to detect genetic interactions in a combined dataset comprising over 2,500 tumors from 23 cancer types. The resulting genetic interaction map of childhood cancers comprises 15 co-occurring and 27 mutually exclusive candidates. The biological explanation of most candidates points to either tumor subtype, pathway epistasis or cooperation while synthetic lethality plays a much smaller role. Thus, other explanations beyond synthetic lethality should be considered when interpreting genetic interaction test results.

[1] Princess Máxima Center for Pediatric Oncology, Heidelberglaan 25, 3584 CG Utrecht, The Netherlands. [2] Computational Biology, St Jude Children's Research Hospital, 262 Danny Thomas Place, Memphis, TN 38105, USA. [3] Hopp Children's Cancer Center Heidelberg (KiTZ) and Division of Pediatric Neurooncology, German Cancer Consortium (DKTK) and German Cancer Research Center (DKFZ), Im Neuenheimer Feld 430, 69120 Heidelberg, Germany. [4] Heidelberg University Hospital, Department of Pediatric Hematology and Oncology, Im Neuenheimer Feld 672, 69120 Heidelberg, Germany. [5] These authors contributed equally: Josephine T. Daub, Saman Amini. ✉email: P.Kemmeren@prinsesmaximacentrum.nl

Cancer is the leading cause of disease-related child death in developed countries, despite increased survival rates from 10% to over 80% in the last decades[1]. Moreover, survival rates differ greatly between different pediatric cancer types and many survivors suffer from severe side effects later on in life[2,3]. Therefore, it is of great importance to gain a better understanding of pediatric cancers and their potential treatments. Pediatric cancers are rare compared to adult cancers and are also fundamentally different. Insights gained from adult cancer studies are thus only for a small part applicable to childhood cancer types. While adult cancers usually occur late in life as the result of a gradual accumulation of somatic mutations, pediatric cancers are rather thought of as developmental diseases. Consequently, they usually have a different cell of origin compared to adult tumors[4]. In addition, they are thought to require a lower number of driver mutations for tumorigenesis and usually exhibit a much lower number of passenger mutations[4]. The exhaustive characterization of genetic alterations has only recently begun in pediatric cancers using whole-genome and exome sequencing approaches[5]. Larger numbers of sequenced tumors for pediatric cancers are now available, allowing the identification of genes that are frequently mutated across cancer genomes[6,7].

Like many other diseases, most cancers do not arise from alterations in individual genes alone but are the result of widespread genetic interactions between them[8]. Genetic interactions occur when the effect of combining two or more alterations in the genome cannot be predicted by adding up the effects of the individual alterations. Genetic interactions are known to be pervasive in model organisms[9–11]. Efforts have also been initiated to map genetic interactions in adult cancer cells[12–15], but have so far been limited in pediatric cancer. One of the primary goals of such studies is to detect synthetic lethal relationships. Synthetic lethality is a type of genetic interaction in which simultaneous disruption of two genes results in cell death[16] while the alteration of only one of the two genes yields a viable cell. For example, cancer cells that harbor mutations in *BRCA1* or *BRCA2* are highly dependent on the function of *PARP1*[17]. Exploiting these types of genetic interactions for therapeutic purposes can therefore advance precision medicine strategies to develop better, patient-tailored, cancer treatments[18,19].

Multiple strategies can be employed to detect synthetic lethal and other genetic interactions in cancer cells (reviewed in ref. [20]). For example, synthetic lethal gene pairs that are identified with high-throughput screens in model organisms such as yeast might indicate synthetic lethality between their human homologs[12]. Another approach is to use RNAi or CRISPR–Cas9 to perform knock-out experiments in human cancer cell lines[13,14]. However, candidate genetic interactions identified with these methods are often hard to validate, as they are context-dependent and might not replicate in a different genomic background, cell type, or cell environment[19].

Another strategy to detect genetic interactions is to perform in silico statistical analyses in large collections of tumor genomes. Applying this approach to adult cancers has led to the identification of co-occurring and mutually exclusive pairs of alterations[21–24]. A pair is co-occurring (or mutually exclusive) when two alterations co-occur more (or less) often than expected by chance. Mutually exclusive altered gene pairs can point to possible synthetic lethal candidates; mutated gene pairs that co-occur often probably co-operate to give the tumor cell a selective advantage. Numerous tools have been developed to identify co-occurring and mutually exclusive interactions between mutated genes in cancer (reviewed in ref. [25]) and differ in the choice of underlying model, statistical approach, and incorporation of existing biological knowledge. However, the interpretation of mutual exclusivity and co-occurrence patterns is an essential but not straightforward step and in silico genetic interaction studies often lack a careful investigation to find biological explanations of their results[25–27].

Here, we draw a comprehensive map of genetic interactions in pediatric cancer. We present a robust pipeline that implements two statistical tests to detect with high confidence significantly co-occurring and mutually exclusive gene pairs. We apply the pipeline on two pediatric cancer data sets, together consisting of over 2500 tumors and 23 cancer types. We perform our analyses both per cancer type and at a pan-cancer level while including the complete set of mutated genes and not restricting to driver genes only. We identify multiple co-occurring and mutually exclusive candidate gene pairs in both data sets. We next investigate potential biological explanations for the patterns of co-occurrence and mutual exclusivity and provide estimates of their contribution to our results. Our findings show that tumor subtype, pathway epistasis, and cooperation are the main biological explanations underlying the results, as each comprises over a quarter of the candidate genetic interactions found. Synthetic lethality plays a much smaller role, as it explains only 7% of the candidate interactions. Taken this outcome we conclude that other explanations beyond synthetic lethality should be considered when interpreting the results of genetic interaction tests.

## Results

**A combined genomic data set covering common pediatric cancers**. To systematically detect genetic interactions in pediatric cancer, a collection of tumor samples from two recently published pediatric cancer data sets was used as a starting point[6,7]. The first data set[6], hereafter called DKFZ, consists of 961 tumors from 23 distinct cancer types (Fig. 1a). The second data set[7], hereafter called TARGET, contains 1699 tumors from six distinct cancer types (Fig. 1b). The combined cohort covers major childhood cancer types including central nervous system tumors, hematological tumors as well as solid tumors. While the DKFZ data set has a focus on central nervous system tumors, the TARGET data set has an emphasis on hematological tumors, thus complementing each other in the extent to which individual tumor types are covered.

Genetic alterations that were considered for the genetic interaction tests include single nucleotide variants (SNVs) as well as small insertions and deletions (indels). Only SNVs and indels in coding regions that likely have a functional effect were used for further analyses. Note that all genes that harbor one or more mutations were included, not limiting the analysis to just the candidate driver genes that are frequently mutated across samples, earlier identified as SMGs[6,7]. The number of mutated genes per individual tumor varies greatly across different pediatric cancer types (Fig. 1c, d) as was also reported before[6,7]. After filtering, a total of 2461 tumors remained in the combined cohort (829 tumors from DKFZ; 1632 tumors from TARGET) and these were used for investigating genetic interactions (see Methods and Supplementary Table 1 for more details).

**A robust statistical pipeline to detect high confidence genetic interactions**. To detect potential genetic interactions that play a role in pediatric cancer, we developed a statistical pipeline to detect pairs of altered genes that co-occur significantly more (or less) often than expected given their individual frequencies (Fig. 2, Methods). A pan-cancer analysis, as well as a test per cancer type to detect cancer-type specific interactions on either the DKFZ or TARGET data set, were performed. As each data set was produced with its own technical and filtering procedures, we applied the tests on each data set separately as merging both sets would possibly lead to false positives. The resulting candidate

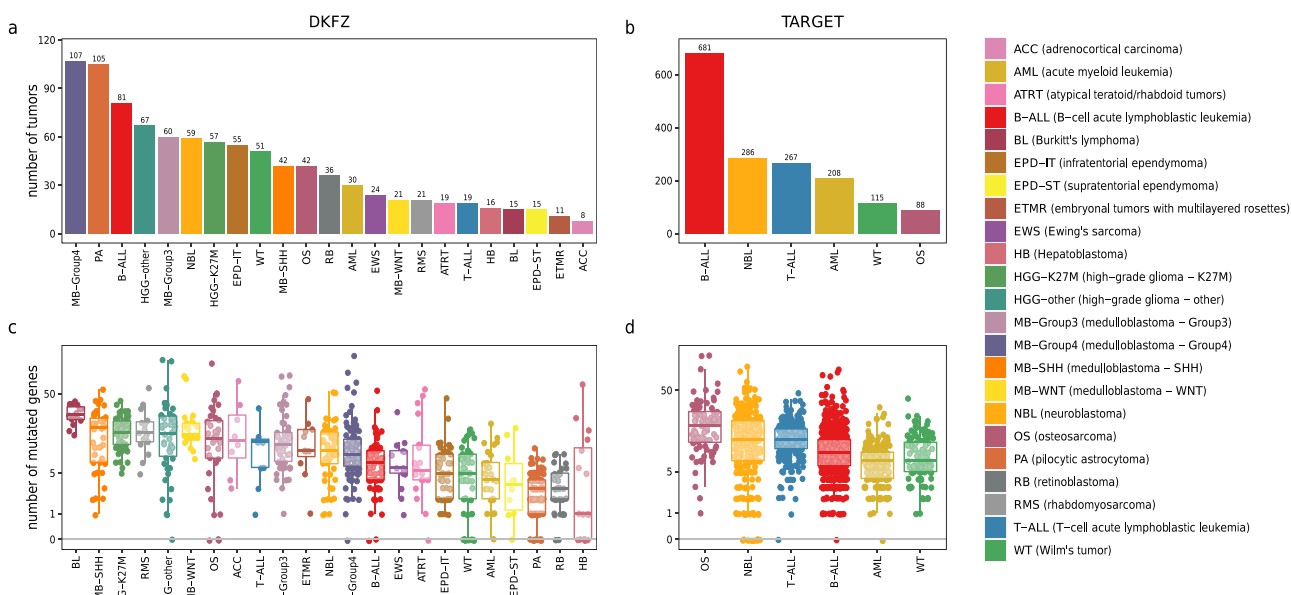

**Fig. 1 Number of samples and mutated genes per cancer type. a**, **b** The number of samples per cancer type in the DKFZ (**a**) and TARGET (**b**) data sets used in this study before sample filtering (e.g., removal of hypermutators and relapse tumors). **c**, **d** Number of mutated genes identified in each individual tumor per cancer type for the DKFZ (**c**) and TARGET (**d**) data sets after sample filtering. Box-plots' center line: median; box limits: upper and lower quartiles; lower (higher) whisker: smallest (largests) observation greater (less) than or equal to lower (higher) box limit—(+) 1.5× interquartile range. Points on the horizontal axis represent tumor samples without any mutated genes. Mutated genes are defined as genes that harbor at least one exonic mutation with a likely functional consequence. Only SNVs and small indels were considered. Source data underlying Fig. 1 can be found in Supplementary Data 2.

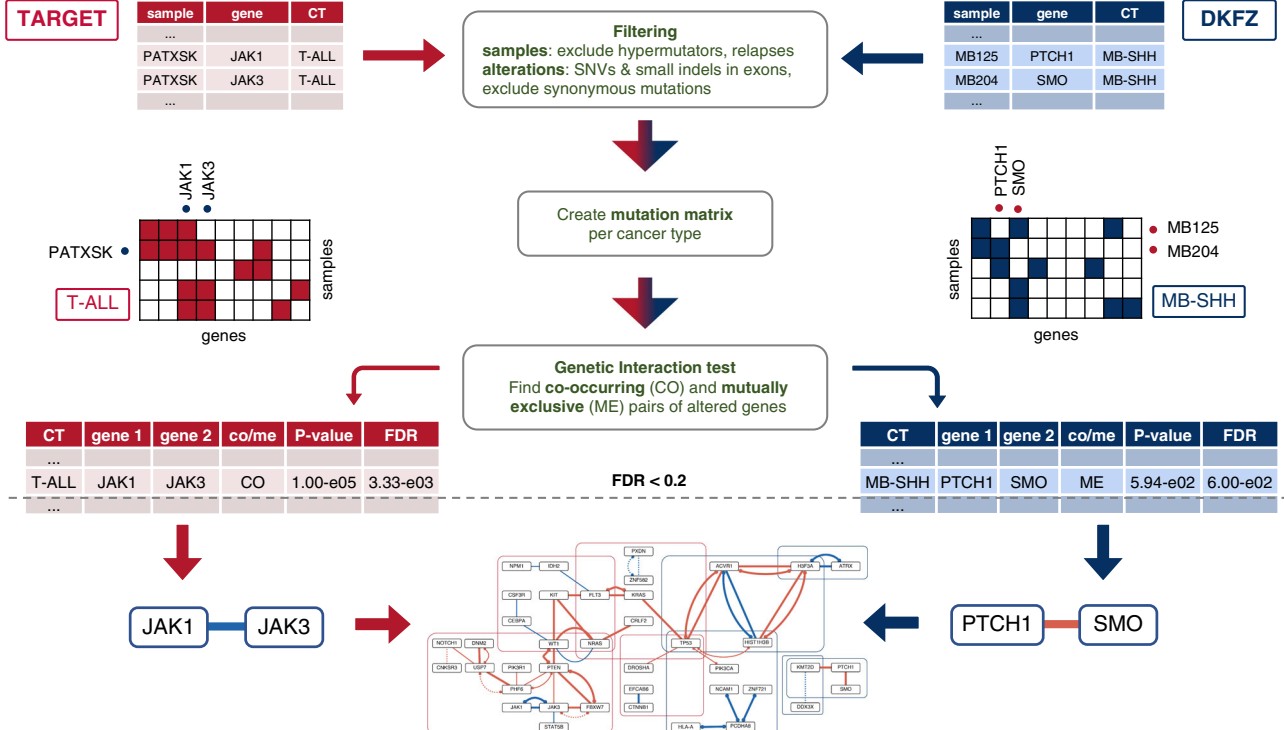

**Fig. 2 Workflow depicting the statistical pipeline to detect genetic interactions.** Starting with the original mutation files from the TARGET and DKFZ data set, samples and alterations are first filtered after which a mutation matrix is created to record which samples contain mutations in a certain gene. Here, the candidate gene pairs JAK1–JAK3 and PTCH1–SMO are shown as examples. The matrix serves as input for the Permutation and WeSME test to detect significantly co-occurring or mutually exclusive pairs of altered genes. The resulting candidates from both data sets are merged into a genetic interaction map of childhood cancer.

gene pairs can point to possible genetic interactions between genes that play an important role in pediatric cancer.

In short, the approach is based on two previously published tests for genetic interactions in cancer. The first test, which we will call hereafter the Permutation test[23] starts with the creation of a binary mutation matrix listing which genes are mutated in which tumor samples. Next, for each gene pair, the number of samples in which both genes are mutated are counted (co-occurrence count). Extreme low values point to mutually exclusive candidate gene pairs, while high counts suggest co-occurring interactions. To infer the significance (P-values) of such extreme counts, a null distribution is created by repeatedly permuting the mutation matrix while keeping its margins fixed. The second genetic interaction test, named WeSME, is based on

the same principle as the Permutation test but has implemented several improvements that significantly speed up the testing procedure while producing highly similar results[28]. P-values resulting from both tests were corrected for multiple testing by estimating an empirical false discovery rate (FDR). Gene pairs scoring an FDR < 0.2 and P-values < 0.1 were considered candidate genetic interactions. To produce high confidence results each WeSME test was repeated ten times (using different randomization seeds) and only those pairs that scored significantly in at least nine out of ten tests were considered as high confidence pairs.

**A comprehensive map of potential genetic interactions in pediatric cancer.** Applying the Permutation and WeSME test on each cancer type in the TARGET data set resulted in a total of 28 candidates divided over the three leukemia cancer types T-acute lymphoblastic leukemia (ALL) (14), acute myeloid leukemia (AML) (8), B-ALL (4), and Wilms tumor (WT) (2) (Table 1 and Fig. 3). In the pan-cancer test, we detected eleven candidates, ten of which were also discovered in the test per cancer type. Of all 29 unique candidate gene pairs, 9 were co-occurring, and 20 were mutually exclusive. Of all candidates, 18 were detected in both the Permutation as well as the WeSME test, 16 in WeSME only, and five in the Permutation test only. Seventeen candidate interactions were discovered earlier in the same data set[7], and twelve candidates had not been detected before. In the DKFZ cancer types, we found a total of eight candidates, all in high grade glioma (HGG) and Medulloblastoma (MB) subtypes, namely HGG-K27M (4), HGG-other (1), MB-SHH (2), and MB-WNT (1) (Table 1 and Fig. 3). Five of these candidates were also found in the pan-cancer analysis, together with five pan-cancer-only candidates. In total, we detected six co-occurring and seven mutually exclusive candidates. Of all candidates, fifteen scored significant in both the Permutation and WeSME test, two in WeSME only and one exclusively in the Permutation test. Only two candidates (ACVR1-HIST1H3B and ATRX-H3F3A) were detected as potential genetic interactions previously in the same data set[6]. Combining the results from both data sets leads to 42

**Table 1 Number of candidate genetic interactions in both data sets.**

|  | TARGET | DKFZ |
|---|---|---|
| *Number of candidates* | | |
| All | 39 | 18 |
| Unique | 29 | 13 |
| *Type of interaction* | | |
| Co-occurring | 9 | 6 |
| Mutually exclusive | 20 | 7 |
| *Which analysis* | | |
| Per cancer type | 28 | 8 |
| PAN | 11 | 10 |
| (overlap cancer type) | (10) | (5) |
| *Previously reported** | | |
| In original paper | 17 | 2 |
| Not reported before | 12 | 11 |
| *Significant in which test* | | |
| Both tests | 18 | 15 |
| WeSME only | 16 | 2 |
| Permutation test only | 5 | 1 |

*Gene pairs that were tested as significantly co-occurring or mutually exclusive in the original papers published accompanying the TARGET[7] and DKFZ[6] data sets.

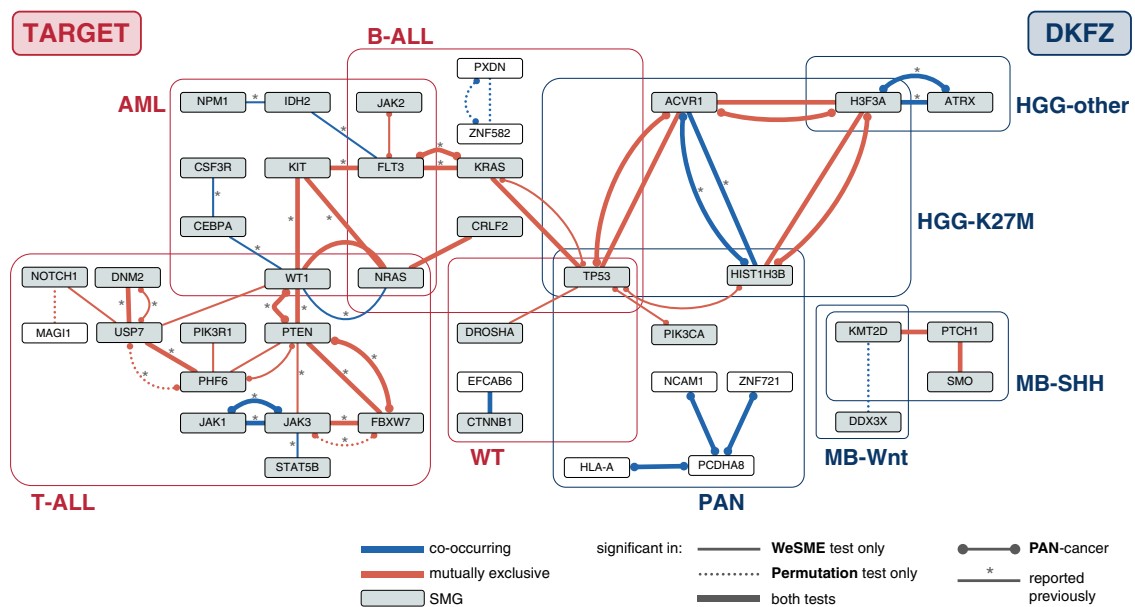

**Fig. 3 Candidate gene pairs resulting from the genetic interaction detection pipeline.** Graph depicting candidate genetic interactions found in either TARGET (red boxes) or DKFZ (blue boxes). Mutually exclusive pairs are indicated in red, co-occurring pairs in blue. SMG significantly mutated genes. "reported previously" refers to gene pairs that were detected as significantly co-occurring or mutually exclusive in the studies accompanying the TARGET[7] and DKFZ[6] data sets. Source data underlying Fig. 3 can be found in Supplementary Data 1.

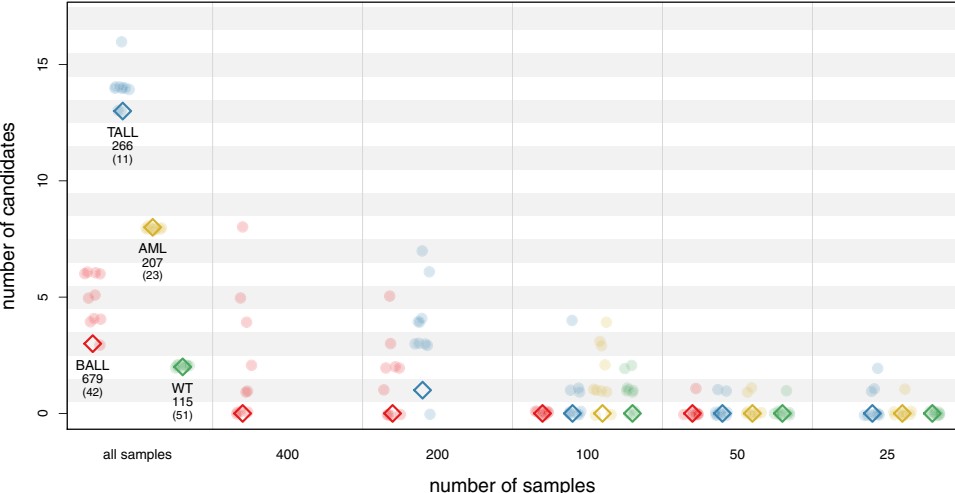

**Fig. 4 Down-sampling the TARGET data set greatly reduces the number of candidates.** Number of candidate pairs for the WeSME test on TARGET cancer types B-ALL, T-ALL, AML, and WT using either all samples, or down-sampled data sets. Dots represent the number of candidates for each of the ten replicates per sample size. Diamonds indicate the number of high confident candidates (being a significant candidate in at least nine out of ten tests). The first category (all samples) shows for each cancer type the original number of samples in the TARGET and -in parentheses- in the DKFZ data set. Source data underlying Fig. 4 can be found in Supplementary Data 2.

candidate genetic interactions, of which 27 mutually exclusive and 15 co-occurring interactions (see Supplementary Data 1 for more details).

**Candidate genetic interactions are highly cancer type specific.** Most genetic interaction candidates that were detected involve possible cancer driver genes, namely genes that were determined to be significantly mutated genes (SMGs) in the previous studies[6,7]. However, in six interactions discovered in this study, non-SMGs were involved, and they are relatively often co-occurring (five candidates) and pan-cancer specific (three candidates). Some of these interactions involve both SMGs and non-SMGs such as for instance CTNNB1 (SMG)–EFCAB6 (non-SMG) in WT and NOTCH1 (SMG)–MAGI1 (non-SMG) in T-ALL. This suggests that the inclusion of the whole set of genes instead of the set of SMGs can lead to new discoveries.

We find about two times more mutually exclusive than co-occurring candidate interactions with our pipeline. The most likely explanation is a technical one. If two mutated genes have a cooperative relationship, but each has a low mutation frequency, the chance of occurring together in the same tumor is even lower and one needs large numbers of samples to have enough power to discover such co-occurring relationships.

The fact that we find less candidate genetic interactions than reported in the earlier two studies on the same data sets has a technical ground as well. Our approach differs in several aspects from the tests applied in the previous studies. First, we include all mutated genes instead of focusing on SMGs only. This way we will detect genetic interactions that involve non-driver genes, but we also introduce more noise which decreases the power of the test. Furthermore, as our pipeline is based on randomizations per cancer type, instead of using, e.g., a Fisher Exact test, we control for the underlying data structure and are more likely to avoid false positives. Third, we applied thresholds on the minimum number of co-occurrences to exclude low-confidence results. Overall, this results in a more conservative estimate of the number of candidate pairs that we predict compared to the previous studies.

Interestingly, none of the discovered gene pairs were found in more than one cancer type albeit several candidates were confirmed in the pan-cancer analysis. Moreover, none of the

candidates were detected in the other data set. Even the genes involved in significant gene pairs appear to be data set specific as only *TP53* is part of candidate interactions in both the TARGET and DKFZ data set.

**Sample size explains lack of overlap in candidates for cancer types present in both data sets.** Even though the six cancer types of the TARGET data set are also present in the DKFZ data set, none of the candidates found in the TARGET data set were confirmed in the DKFZ analysis. The most probable explanation would be the fact that the number of samples of these cancer types is much lower in the DKFZ set compared to the TARGET set. Indeed, one of the important requirements to detect genetic interactions in silico, is to test large numbers of samples to have sufficient statistical power. The number of samples is usually not a limiting factor for adult cancers, but sample size can be an issue in rare diseases such as pediatric cancers.

To investigate if sample size indeed is the underlying cause, we performed a down-sampling analysis where we performed the WeSME test on random subsamples of the four cancer types for which we detected candidates in the TARGET data set, namely B-ALL, T-ALL, AML, and Wilm's tumor (WT). As shown in Fig. 4, we observe a wide variation in the number of candidates, especially when testing larger sample sizes (200–400). Repeating the WeSME test ten times and only considering candidates that score significantly in at least nine out of ten tests thus proves to be a good strategy to detect high-confidence candidates. More importantly, the number of discoveries drops dramatically with sample sizes of 100 and below, which explains the lack of candidates in these four cancer types in the DKFZ data set, as their number of samples lies in the range between 11 and 51.

**Tumor subtype rather than mutation load association underlies most T-ALL candidates.** Recently it has been shown by Van de Haar and colleagues that tumor subtype and tumor mutation load can be confounding factors in tests to detect genetic interactions in cancer data sets[27]. When the mutation frequency of a gene is strongly associated with the tumor mutation load of a sample, such a gene will likely turn up as a mutually exclusive candidate with mutated genes that have a negative association with tumor mutation load. We investigated whether tumor

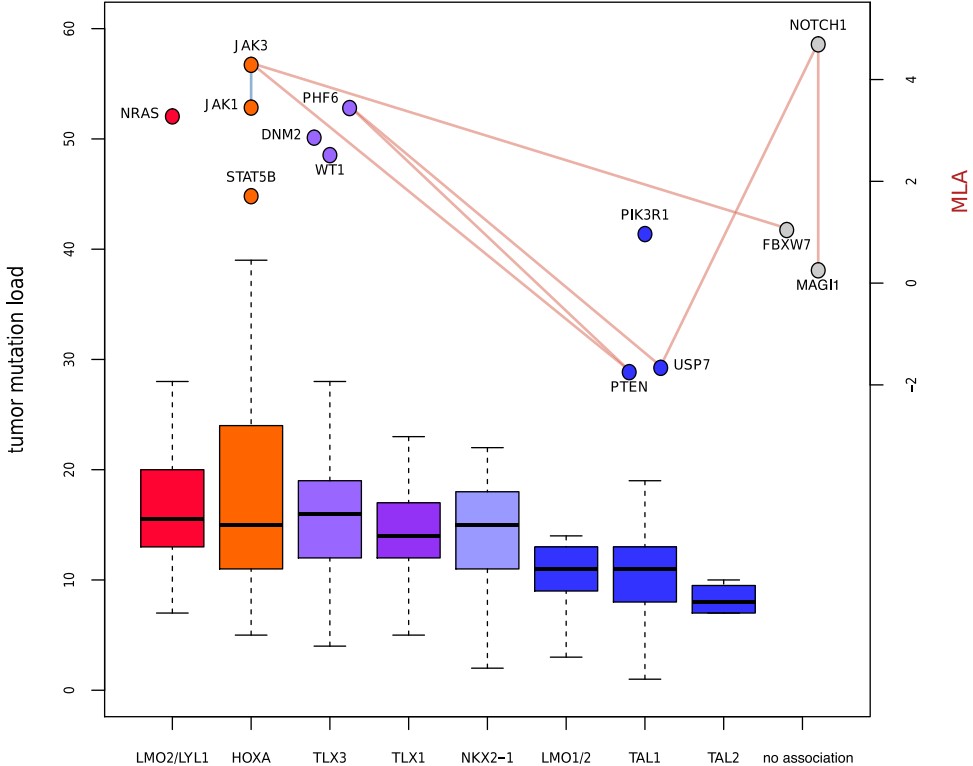

**Fig. 5 Tumor mutation load is lower in more mature T-ALL subtypes.** Sample tumor mutation load for each T-all subtype, with subtypes ordered (left to right) from early to late T-cell progenitor stage. Genes that were part of a candidate genetic interaction and that were significantly associated with a T-ALL subtype are depicted in the figure with their corresponding subtype and their MLA score (secondary vertical axis). Subtype ordering, coloring and assignment to genes were reproduced from Fig. 3 in ref. [31]. Edges between genes represent "suspect" candidate genetic interactions found in T-ALL, defined as mutually exclusive gene pairs (shown in red) with one gene having an MLA score > 3 and together having a difference in MLA score > 3 or a co-occurring pair (shown in blue) with both genes having an MLA > 3. Source data underlying Fig. 5 can be found in Supplementary Data 2.

mutation load can explain part of our candidate genetic interaction pairs. We calculated the tumor mutation load per sample and the mutation load association (MLA[27]) of each candidate gene, where MLA is an indication of the association between the mutation load and likelihood of being mutated (Methods). Compared to the Van de Haar study, which found MLA scores between minus four and ten with a median of five, we find in general much lower MLA values per gene (Supplementary Figs. 1 and 2), indicating that overall there is no strong correlation with tumor mutation load in pediatric cancer. This most likely can be explained by the much smaller number of passenger mutations in pediatric cancers.

Next, we defined as "suspect" all candidate mutually exclusive gene pairs with an MLA difference larger than three and at least one gene having an MLA larger than three or with both genes having an MLA larger than three in case of co-occurrence. We found seven of such suspect candidates, all belonging to T-ALL (Fig. 5, Supplementary Table 2, Supplementary Fig. 3). Most T-ALL tumors can be assigned to different subtypes, characterized by mutually exclusive structural aberrations and subtype-specific expression profiles[29–31]. More specifically, T-ALL subtypes reflect the maturation state of the T-cell progenitor from which the tumor developed. Since we do have subtype annotation available for almost all available T-ALL samples from the TARGET data set ("Methods"), we examined whether there is an association between tumor subtype and tumor mutation load. Indeed, there is a clear difference between subtypes, with a tendency of tumor types associated with primitive T-cell progenitors carrying more mutations than subtypes associated

with more differentiated precursors (Fig. 5). Four of the seven suspects mutually exclusive candidate T-ALL gene pairs can likely be explained by the fact that mutated genes in these pairs are enriched in different subtypes with different mutation loads (Supplementary Table 2). The remaining three gene pairs involve one of the commonly mutated genes *NOTCH1* and *FBXW7* and could be false positives (Supplementary Fig. 3), but are likely the result of other biological processes.

**Four biological explanations underly most candidate gene pairs.** Next, we extensively investigated for each of the candidate genetic interactions which underlying biology could explain their mutually exclusive or co-occurring relationship. This entailed consulting experts in the field of the corresponding cancer types as well as an extensive literature search. We considered the function of the individual genes, known relationships between the found gene pairs, and their role in cancer, with specific focus on the cancer type in which the candidate was detected. In addition, we developed a Candidate Reporting Tool, which is an R shiny web application (https://gi-analysis.kemmerenlab.eu/) to aid in the inspection of mutational information on each candidate gene pair, producing multiple visualizations together with a table listing the biological or pathogenic consequence of each individual mutation in the corresponding genes. We have listed all candidates in Table 2 together with the most likely biological explanation for the detected mutual exclusivity or co-occurrence. Supplementary Note 1 contains detailed explanations and literature used for all candidates.

**Table 2 Candidate genetic interactions and their underlying explanation.**

| Cancer type | Gene1 | Gene2 | co/me | Most likely explanation | Reference(s) |
|---|---|---|---|---|---|
| *DKFZ* | | | | | |
| HGG-K27M | ACVR1 | HIST1H3B | co * | Cooperation | 49 |
| | ACVR1 | H3F3A | me * | Subtype | 32,33 |
| | ACVR1 | TP53 | me * | | |
| | H3F3A | HIST1H3B | me * | | |
| HGG-other | ATRX | H3F3A | co * | Cooperation | 39,50,51 |
| MB-SHH | KMT2D | PTCH1 | me | Pathway epistasis | 52,53 |
| | PTCH1 | SMO | me | Pathway epistasis | 36 |
| MB-WNT | DDX3X | KMT2D | co | Cooperation | 54 |
| PAN | HLA-A | PCDHA8 | co | False positives | |
| | NCAM1 | PCDHA8 | co | | |
| | PCDHA8 | ZNF721 | co | | |
| | HIST1H3B | TP53 | me | Subtype | 32,33 |
| | PIK3CA | TP53 | me | Pathway epistasis | 55 |
| *TARGET* | | | | | |
| AML | CEBPA | CSF3R | co | Cooperation | 56,57 |
| | CEBPA | WT1 | co | Cooperation | 58,59 |
| | FLT3 | IDH2 | co | Cooperation | 60–62 |
| | IDH2 | NPM1 | co | Cooperation | 61 |
| | FLT3 | KIT | me | Pathway epistasis | 63 |
| | KIT | NRAS | me | Pathway epistasis | 37 |
| | KIT | WT1 | me | Synth lethality | 37 |
| | NRAS | WT1 | me | Synth lethality | 37,64 |
| B-ALL | PXDN | ZNF582 | co * | False positive | |
| | CRLF2 | NRAS | me | Subtype | 65 |
| | FLT3 | KRAS | me * | Pathway epistasis | 63,66,67 |
| | KRAS | TP53 | me * | Subtype | 65 |
| T-ALL | JAK1 | JAK3 | co * | Cooperation | 68 |
| | JAK3 | STAT5B | co | Cooperation | 69,70 |
| | NRAS | WT1 | co | Cooperation | 71 |
| | DNM2 | USP7 | me * | Subtype | 31 |
| | FBXW7 | JAK3 | me * | Pathway epistasis | 29,72,73 |
| | FBXW7 | PTEN | me * | Pathway epistasis | 29,72 |
| | JAK3 | PTEN | me | Pathway epistasis | 73 |
| | MAGI1 | NOTCH1 | me | Pathway epistasis | 72,74 |
| | NOTCH1 | USP7 | me | Synthetic lethality | 75 |
| | PHF6 | PIK3R1 | me | Subtype | 30,31 |
| | PHF6 | PTEN | me * | Subtype | 30,31 |
| | PHF6 | USP7 | me * | Subtype | 31 |
| | PTEN | WT1 | me * | Pathway epistasis | 30,71,73 |
| | USP7 | WT1 | me | Subtype | 31 |
| WT | CTNNB1 | EFCAB6 | co | Cooperation | 76 |
| | DROSHA | TP53 | me | Subtype | 77 |
| PAN | FLT3 | JAK2 | me | Pathway epistasis | 65,78,79 |

*co* co-occurrence, *me* mutual exclusivity.
*Also significant in PAN cancer test.

As indicated before, many of the candidate gene pairs have a co-occurring or mutually exclusive relationship because the cancer type in which they were detected actually consists of multiple subtypes. Apart from the T-all examples described above, this is for example the case for the high-grade glioma K27M group. The mutually exclusive candidate gene-pair HIST1H3B-H3F3A in this group can be explained by the fact that patients with this particular cancer type can be divided into two subgroups, defined by K27M mutations in either the HIST1H3B or H3F3A genes that code for histones H3.1 and H3.3, respectively[32–34]. Typically, patients in the H3.1 subgroup also carry ACVR1 mutations, explaining the ACVR1-HIST1H3B co-occurrence and ACVR1-H3F3A mutual exclusivity relationships. Similarly, TP53 is almost exclusively mutated in the H3.3 group, again explaining mutually exclusivity between TP53 and ACVR1. All these candidates, together with the mutually exclusive TP53-HIST1H3B pair, were also found in the PAN cancer test,

due to this strong signal in HGG-K27M, rather than similar patterns in other cancer types.

Other candidate gene pairs can be explained by pathway epistasis (or functional redundancy), the phenomenon in which the mutation of one gene will have the same effect on the activation or inactivation of a pathway as mutating another (downstream) gene. In such cases, mutating either gene will result in the same effect, and therefore one mutation is enough for the tumor cell to gain a selective advantage. For example, mutations in PTCH1 and SMO were found to be mutually exclusive in the MB SHH cancer type. Both genes are part of the Sonic Hedgehog (Shh) pathway, which is assumed to have a tumor driver role in this type of MB[35]. PTCH1 normally inhibits the activation of SMO, but the presence of Hedgehog (Shh) leads to suppression of PTCH1 and consecutive activation of the Shh pathway by SMO. It has been shown that SMO inhibitors are most effective in MB-SHH patients with SMO or PTCH1 mutations[36], suggesting that

*SMO* mutations found in MB-SHH samples do not have a knockout effect on the gene, but more likely release or circumvent its inhibition by PTCH1. This likely explains the mutually exclusive relationship found between altered *PTCH1* and *SMO* genes, as both have the same effect and therefore are redundant mutations.

Mutual exclusivity could also point to a possible synthetic lethal relationship between genes, where mutating either of the two genes has no effect on cell viability but mutating both genes will result in cell death. We found several candidate interactions that could be driven by synthetic lethality, although no previous studies were found that confirm such a relationship. One such gene pair is *NRAS-WT1*, found to be mutually exclusive in AML. *NRAS* is, together with *KRAS*, part of the GTPase family of genes that are often activated in cancer. It has been shown that in *KRAS*-dependent tumors, inactivation of *WT1* will reduce tumor formation[37]. If this effect also applies to *NRAS* mutated tumors, *NRAS-WT1* could be a synthetically lethal gene pair. RAS genes act downstream of *KIT* in the RTK signaling pathway, therefore the synthetically lethal relationship between *KRAS/NRAS* and *WT1* might also translate to the mutually exclusive *KIT-WT1* candidate in AML. Interestingly, the gene-pair *NRAS-WT1* is a co-occurring candidate in T-ALL. *WT1* is known to play different roles in cancer, acting both as a tumor suppressor and oncogene, where its function often depends on which isoform is expressed[38]. The apparent adverse interaction between *NRAS* and *WT1* in AML and T-ALL might be due to these different roles of WT1 in cancer development and warrants further investigation.

Candidate co-occurring gene pairs possibly point to a cooperative relationship, in which the mutation in one gene reinforces the effect of mutating the other gene. An example of such a gene pair is *ATRX-H3F3A* in the HGG Other group. The *H3F3A* mutations in this group are not K27M, but G34R/V H3 mutations, another recurrent *H3F3A* alteration in HGGs. *ATRX* regulates chromatin remodeling and transcription and plays an important role in maintaining genome stability by recruiting H3.3 at telomeres and pericentric heterochromatin. It has been suggested that the loss of *ATRX* could prevent mutant H3.3 from altering transcription of specific oncogenes[39], which would suggest a co-operative role of both altered genes in tumorigenesis.

Considering the extensive literature search performed for all candidate genetic interactions from both data sets (Table 2, Supplementary Note 1), we found that cancer subtype (12 gene pairs) is the most likely underlying cause in about 44% of the mutually exclusive candidates and 29% of all candidates. Pathway epistasis (12 pairs) accounts for another 44% of the mutually exclusive candidates (29% of all candidates) and 11% (3 pairs) may be caused by synthetic lethality (7% of all candidates). About 73% of co-occurrences (11 pairs, 26% of all candidates) could be attributed to co-operation while the remaining 27% (4 pairs, 10% of all candidates) are likely false positives.

## Discussion

Here, we built a map of genetic interactions in childhood cancer. We applied a robust statistical pipeline and detected 27 mutually exclusive and 15 co-occurring altered gene pairs across two pediatric pan cancer data sets. As we rely on the frequency of mutated genes occurring in individual cancer types, underlying phenomena such as cancer heterogeneity, low sample numbers or low mutational burden limit the power to detect genetic interactions. This might in part also explain the relatively low number of co-occurrences and mutually exclusive gene pairs detected here. We implemented two related methods for genetic interaction detection: the permutation test and the WeSME test. Most of the candidate gene pairs were detected with both tests, confirming

that the faster WeSME test is a good alternative for the permutation test. Indeed, we find a strong correlation between the *P*-values of all tested gene pairs that were produced by both tests (Pearson's $r = 0.97$, *P*-value $< 2.2e-16$). Note however that the WeSME test has a slight tendency toward lower mutual exclusivity *P*-values, while the Permutation test yields lower co-occurrence *P*-values (Supplementary Fig. 4) as was shown earlier by Kim et al.[28]. In addition, a large majority (77%) of the cancer type-specific candidate interactions could be reproduced with DISCOVER, an alternative genetic interaction test that was shown to be more sensitive than various other methods while controlling for false positives[26]. Out of 36 cancer type specific candidates, 26 scored an FDR < 20% in DISCOVER with two additional pairs scoring an FDR < 25% (Supplementary Data 1). Note that DISCOVER only allows for cancer type-specific testing, and therefore we could not validate our PAN cancer results (six candidate pairs) with this method.

It has been previously shown that genetic interactions are tissue and cancer type specific[23]. Indeed, we confirm this finding in our current study as no candidate genetic interaction was detected in more than one cancer type. This is partly because apart from a few commonly mutated genes in various cancers, most cancer types are characterized by recurring alterations in specific genes, showing that the process of oncogenesis is unique within each tumor (sub) type. The sensitivity to certain genetic hits at a specific maturation state of the cell of origin is a main driver underlying this process. As an additional consequence, we have more power to find genetic interactions between these frequently mutated genes, compared to rarely mutated genes.

Although all cancer types in the TARGET data set are also part of the DKFZ set, we did not reproduce the TARGET candidates when testing the same cancer types in DKFZ. We showed with downsampling that the most likely explanation for this lack of overlap is the reduced samples size for these cancer types in DKFZ. This shows the limited power of the current study with few samples for many of the cancer types and stresses the importance of having a larger data set across all cancer types to detect genetic interactions. Especially in pediatric cancer this poses a challenge as most cancer types are rare and data sets produced within one institute are relatively small. Current initiatives to collaborate between institutes and countries to build large (pediatric) cancer resources consisting of combined genomic data sets is a key step towards a solution. Examples of such collaborations are the GENIE project[40], the Children's Brain Tumor Tissue Consortium[41] and St. Jude Cloud[42]. However, merging data sets imposes new challenges, since the application of different protocols, sequencing techniques, and quality filtering methods will introduce confounding factors leading to batch effects and consequently false positives. Future genetic interaction tests applied on merged data sets should therefore take the underlying heterogeneity of these different sources into account to overcome the limitations of using tests on smaller individual data sets as applied here. This could be performed by adjusting the approach presented here for the PAN cancer test, by counting co-occurrences and mutual exclusivity over the whole set, while applying the randomizations within cancer types. This approach could be easily translated to a framework in which randomizations take place among samples from the same source, while summary statistics are collected over the combined data set.

For practical utility in mechanism-of-action based therapies as well as for understanding the biology driving cancer progression, we need a much better understanding of the causes underlying genetic interactions. Here, we have only taken a first step towards a thorough understanding of the underlying biology by grouping the potential genetic interactions pairs in different biological explanations based on existing literature. No functional validation

has however been performed in this study, which is key to decisively determine the biological explanation for many of the pairs found. The biological explanations indicated here should therefore all be considered as highly likely but warrant further follow-up experimentation to confirm them. Mutual exclusivity found between altered gene pairs is often the result of mutated genes being specific for different cancer subtypes. For the same reason co-occurrence will be detected between mutated genes that are enriched in certain cancer subtypes. This underscores the importance to distinguish between subtypes in the clinic because it is likely that they will respond differently to the same treatment. Co-occurrence could also indicate possible cooperation between both genes in cancer development. Finding such cooperating pairs will give us more insight in the pathways involved in tumorigenesis. Pathway epistasis is another source of mutual exclusivity, as mutating genes that are part of the same pathway may have the same downstream effect, and as a result having only one gene mutated is sufficient to have a beneficial effect for the tumor. Mutual exclusivity or co-occurrence could also occur as a result of MLA bias, leading to false-positive candidates. Indeed, we found several mutually exclusive candidate gene pairs in T-ALL with large MLA differences and one co-occurring pair of high MLA genes. Most of these "suspect" gene pairs are however more likely explained by the fact that the genes involved are associated with different subtypes. Interestingly, we also showed that in T-ALL mutation load decreases with the maturation state of the tumor cell. Synthetic lethality is another explanation for mutual exclusivity. Synthetic lethality is of particular importance for cancer research because it can be applied as a strategy in targeted cancer treatment, where pharmaceutical inhibition of one gene combined with a mutation in its synthetic lethal partner gene in the tumor induces cancer cell death, while sparing healthy cells. An example of this is the FDA approved application of PARP inhibitors to treat breast cancer patients with a BRCA1 or BRCA2 mutation[8]. (Note, this gene pair would likely not have been detected with our method since it is based on induced essentiality: PARP mutations alone are not beneficial for the tumor and only become essential after mutation of BRCA1/2). Progress in other areas is however limited, often due to a poor understanding of the underlying mechanism, warranting further efforts to invest more in not only finding genetic interactions, but also understanding the biology behind them[43].

Taken together, we show in our first map of genetic interactions in childhood cancer that cancer subtype, pathway epistasis, and cooperation are the main underlying biological explanations for our candidates, likely contributing each around 26–29% of all gene pairs. Only in 7% of the candidate gene pairs, synthetic lethality is the most likely explanation, while another 10% were most likely false positives due to technical artifacts. These percentages differ from expectations in most genetic interaction tests. Most tests aim to detect patterns of pathway epistasis or synthetic lethality in cancer but overlook the fact that other explanations play at least an equally important role. Studies into genetic interactions in cancer should therefore not stop at producing lists of candidate gene pairs but need to investigate what the most likely explanations are before drawing conclusions. The map also makes clear that these findings are only the first steps towards exploring the full spectrum of genetic interactions in pediatric cancer and much more data and functional validation is needed to understand the complete picture.

## Methods

**Data collection and processing**. To systematically detect genetic interactions in pediatric cancers, a collection of tumor samples from two recently published pediatric cancer studies was used in our pipeline. One study[6] was carried out by the German Cancer Research Center (Deutsche Krebsforschungszentrum, DKFZ)

while the second study is part of the Therapeutically Applicable Research to Generate Effective Treatments (TARGET) initiative[7]. Informed consent has been obtained for all subjects involved in both original studies at the time of sample collection through the corresponding informed consent procedures and protocols. Approval for use of the subject's data within the context of this study has been granted by the Data Access Committees of the TARGET and DKFZ datasets.

**DKFZ data set**. The DKFZ cohort covers 24 major childhood cancer types with an emphasis on central nervous system tumors (including atypical teratoid/rhabdoid tumors; embryonal tumors with multilayered rosettes; the four MB groups WNT (MB-WNT), SSH (MB-SSH), Group3 (MB-GR3) and Group4 (MB-GR4), pilocytic astrocytoma; high-grade glioma with and without histone 3 K27M mutations (HGG-K27M, HGG-other); infratentorial and supratentorial ependymoma (EPD-IT, EPD-ST)), hematological tumors (AML; B-cell ALL with or without hypodiploidy (B-ALL-HYPO, B-ALL-other); T-cell ALL (T-ALL); Burkitt's lymphoma) as well as solid tumors (neuroblastoma (NB); WT, osteosarcoma (OS); Ewing's sarcoma (EWS); hepatoblastoma (HB); adrenocortical carcinoma (ACT); rhabdomyosarcoma and RB). Note that to be able to compare results from both DKFZ and TARGET data sets, we merged B-ALL-HYPO and B-ALL-other samples into one B-ALL group.

The data set consists of 961 tumors (from 914 patients) sequenced with both paired and single end Illumina-based technology, including whole-genome sequences (WGS) and whole-exome sequences (WES). To avoid false positives, 82 relapse samples were excluded, as well as seven hypermutator samples (>ten coding mutations per Mb, see paragraph Hypermutator filtering) from the HGG-other cancer type (ICGC_GBM15, ICGC_GBM56, ICGC_GBM6, ICGC_GBM67, SJHGG030, SJHGG034, SJHGG111), and ten single-end sequenced samples (MB_Exm250, MB_Exm528, MB_Exm10, MB_Exm564, MB_Exm1001, MB_Exm1017, MB_Exm17, MB_Exm516, MB_Exm575, and MB_Exm879).

Mutation data are available and can be downloaded from publicly available data portals such as http://pedpancan.com, but are limited to frequently mutated genes. In this study, we aim to include all mutated genes and therefore use the high confident, but unfiltered results of the variant calling procedure described in Gröbner et al.[6]. In summary, raw FASTQ files were processed by the standardized alignment and variant calling pipeline developed by and applied in the ICGC Pan-Cancer project (https://github.com/ICGC-TCGA-PanCancer). The human genome assembly hs37d5 (ncbi.nlm.nih.gov/assembly/2758) was used as a reference genome and GENCODE19 (gencodegenes.org/releases/19.html) for gene annotation. Germline variants were determined based on their presence in the matched control tissue.

In cases where mutations were annotated to multiple genes, one of these genes was selected using a voting system based on annotation fields derived from Gencode version 19 (v19, used by DKFZ in their variant calling pipeline) and Gencode version 27 (v27, the most recent version at the time of doing the analyses). In this voting system, the first ranking gene was chosen after sorting the genes on the following properties: Gencode v19 status (KNOWN, NOVEL, and PUTATIVE), Gencode v19 type (protein-coding, other), Gencode v27 type (protein-coding, other), the total number of exonic alterations for this gene (higher numbers ranking higher), the total number of exonic alterations in single genes (so alterations not overlapping other genes ranking higher) and gene name not containing "-" (which usually indicates a read-through gene).

Only somatic SNVs and small insertions and deletions (indels) were selected for this study and only those located in exonic regions of protein-coding genes that might have a functional consequence (frameshift, non-frameshift, non-synonymous, stopgain, and stoploss) according to ANNOVAR annotation were included. In total, 9922 SNVs and 1236 indels remained for downstream analysis. Genes with at least one of the abovementioned mutations were considered as "mutated genes" and were kept for each sample. The final DKFZ data set that was used to identify genetic interactions includes 829 tumors, comprising 523 WGS and 306 WES samples. See Supplementary Table 1 for an overview of the consequences of each filtering step on tumor numbers in both data sets.

**TARGET data set**. The TARGET data set comprises of 1699 primary tumors, of which 1648 tumors with WGS or WES data (from 655 and 1115 samples respectively) were available for download. Three tumors (10-PARTJJ, 10-PAN-WIM, and 30-PARJXH) were excluded as they could not be linked with corresponding metadata files. The remaining data set consists of six cancer types with most samples coming from hematological tumors (AML, 208); B-ALL, 681; T-ALL, 267) as well as solid tumors (NB, 286); OS, 88; WT, 115). MAF files with annotated somatic SNVs and indels for these cancer types were downloaded on March 21, 2018 (https://ocg.cancer.gov/programs/target/data-matrix). The variants listed in these files are the result of an initial variant calling pipeline using whole-exome sequencing and Complete Genomics Inc. whole genome sequencing technology followed by thorough filtering as described in Ma et al.[7].

Applying the same criteria as with the DKFZ set for hypermutators, one B-ALL tumor (10-PARBPX) and one NB tumor (30-PAPPKJ) were excluded from further analyses. From the remaining 1643 tumors a total of 19,865 SNVs and 2272 indels located in exons and likely having a functional effect (annotated as Missense_Mutation, Nonsense_Mutation, missense, nonsense, Frame_Shift_Del, Frame_Shift_Ins, In_Frame_Del, In_Frame_Ins, frameshift, proteinDel,

proteinIns) were used to detect genetic interactions. Eleven tumors did not contain such functional alterations, resulting in a total of 1632 tumors (649 WGS samples and 1105 WES samples) that were included in downstream analyses.

**Hypermutator filtering**. Hypermutators are usually defined as tumors with more than ten coding mutations per Mb, but there is currently no consensus about what type of mutations and which genomic region should be included to define a hypermutator[44]. In this study, where we test two data sets from different sources, we decided to apply the threshold of ten mutations per Mb on a strictly defined coding region of known length and counting the number of mutations in this region including all SNVs (missense, nonsense, and silent) and all indels. We used the GeneBase tool[45] to extract the non-redundant length of the coding part of exons in a set of protein-coding genes from the NCBI database where the gene and its transcripts have status REVIEWED/VALIDATED. The exact non-redundant length of the coding part of these 18,255 genes is 23,698,355 bp (23.7 Mb). Assuming that the sequencing reads in our data sets have at least 95% sufficient coverage in this region, so having 22.5 Mb or more covered, each sample with more than 225 coding mutations in this set of genes will be considered a hypermutator.

### Genetic interaction analyses, statistics, and reproducibility

*Permutation test*. To detect significant cases of co-occurrence and mutual exclusivity, we followed a permutation approach similar as described in[23]. First, for each cancer type, we constructed binary mutation matrices, recording per sample if it has one or more mutations (SNVs or indels) in a certain gene (Supplementary Fig. 5). Genes with only one mutation in the mutation matrix were excluded to reduce computational time. For each gene pair we counted the number of co-occurrences, that is the number of samples that have both genes mutated. Secondly, we created a null distribution of co-occurrence counts using the *Permatswap* function in the *vegan* R library (version 2.5–4, default parameters, except for $mtype = $ "prab") to generate a series of permuted matrices ($N = 1 \times 10^6$), while keeping their margins fixed. In other words, with each permutation, the total number of mutated genes per gene and per sample still match with the original mutation matrix. This way, the null distribution reflects the underlying heterogeneity (variability in gene- and tumor-level mutation rates) of the data set. Next, an empirical *P*-value for co-occurrence ($P_{co}$) was calculated for each pair of mutated genes by taking the proportion of permutations in which the co-occurrence count was equal to or higher than the observed count. To be more specific, the $P_{co}$ value for a gene pair ($g_1, g_2$) was calculated as follows:

$$P_{co}(g_1, g_2) = \frac{1 + \sum_{i=1}^{N} \left[ co_i(g_1, g_2) \geq co_{obs}(g_1, g_2) \right]}{1 + N} \quad (1)$$

where $N$ is the number of permutations and $co_{obs}(g_1, g_2)$ and $co_i(g_1, g_2)$ represent the co-occurrence count for genes $g_1$ and $g_2$ in the observed and $i$th permuted matrix respectively. Note that we add a one in both the numerator and denominator to have a good estimator of the *P*-value and to avoid *P*-values of zero[46]. In a similar fashion, a *P*-value for mutual exclusivity ($P_{me}$) was calculated as the proportion of permutations in which the co-occurrence count was equal to or lower than the observed data.

As the empirical *P*-values were not uniformly distributed and showed a bias towards one, we could not use standard FDR calculations to correct for multiple hypothesis testing. Instead, we estimated the FDR empirically by creating a *P*-value null distribution. We randomly selected 100 matrices from the permuted matrices and performed the genetic interaction permutation test on each matrix to generate a random set of *P*-values, $S_{null}$ (Fig detailed workflow test). For each observed *P*-value $P^*$ (computed in the previous step), the FDR was estimated as follows (if all hypotheses with a *P*-value $\leq P^*$ would be rejected):

$$FDR\left(P^*\right) = \frac{V(P^*)}{R(P^*)} \quad (2)$$

Where $V(P^*)$ is the estimated number of false positives and $R(P^*)$ is the total number of rejected null hypotheses. $V(P^*)$ was estimated from the proportion of *P*-values $\leq P^*$ in $S_{null}$ multiplied by the total number of observed *P*-values. Finally, a *Q*-value was determined by taking the lowest estimated FDR among all observed *P*-values $\geq P^*$. Gene pairs were considered as significant if they scored a *Q*-value < 0.2 and a *P*-value < 0.1 and had a minimum co-occurrence count of three when testing for co-occurring gene pairs.

In the pan-cancer analysis, we performed the same test on a mutation matrix constructed with all cancer types combined, but permutations were carried out for each cancer type separately to control for any biases in mutation frequencies.

*WeSME test*. As the matrix permutation approach is rather time-consuming, we also applied a faster genetic interaction test and compared the results of both tests to infer their overlap. This test, called WeSME, starts similar to the permutation approach with a gene-sample mutation matrix from which for each gene pair the number of mutual exclusive samples is counted[28]. Instead of permuting this matrix many times to compute a *P*-value, WeSME uses a weighted sampling approach based on the mutation rate of the samples. The method also reduces computation

time by restricting the number of resamplings as it starts with a small null distribution and only increases the number of resamplings (to a maximum of $N = 10,000$) for candidate gene pairs, namely those that have a low *P*-value estimated from the initial null distribution.

To infer the FDR, a similar empirical FDR approach is used as with the matrix permutation method, by permuting the mutation matrix 300 times and applying the WeSME test on the permuted matrices to create a *P*-value null distribution. WeSME places genes in either of two mutation rate bins "high" and "low" (with 2% of the samples being mutated as threshold) and compares the observed *P*-values with a *P*-value distribution of gene pairs from similar bins. The null *P*-values are thus split into three distributions representing the combinations of mutation rate bins ([low,low], [low, high] and [high,high]). Since null *P*-values are assigned to three different bins, 300 permutations were performed instead of 100 as was done in the permutation test. The WeSME Python scripts were downloaded from https://www.ncbi.nlm.nih.gov/CBBresearch/Przytycka/index.cgi#wesme and modified to make it suitable for the current study. In particular, adaptions to the original scripts were made to run a pan-cancer analysis by pooling data from multiple cancer types, counting mutual exclusivity over the pooled set, but performing resampling within each cancer type. To reduce computational time in the PAN cancer test, gene pairs were only tested for mutual exclusivity if the total number of mutually exclusive samples was at least three. With the co-occurrence test, at least one sample needed to have a co-occurring mutated gene pair. In contrast to the original WeSME code, the *P*-values of all gene pairs were used for FDR calculations instead of only keeping gene pairs with *P*-values below 0.1. As in the Permutation test, only gene pairs with an empirical FDR < 0.2, a *P*-value < 0.1, and a minimum co-occurrence count of three (only when testing for co-occurring gene pairs) were considered as significant candidates for further analysis.

To avoid potential biases due to this sampling-based method, we repeated the analyses ten times, each time using another randomization seed, and considered only as high confidence candidates those gene pairs that scored significant in at least nine out of ten tests. Note that the Permutation Test is also based on resampling but was only run once because of its higher computation time.

**Mutation load analyses**. For each gene that was part of a candidate gene pair, the MLA was calculated following[27]. In short, we used the R function glm from the stats package to run a logistic regression of mutation frequency on mutation load and calculated the MLA score by dividing the regression coefficient by the standard error. Both the mutation frequency (of a gene) and mutation load (of a sample) were calculated from the column and row sums respectively of the sample-gene mutation matrix used in our genetic interaction test. All candidate gene pairs with an MLA difference larger than three and at least one gene with an MLA larger than three were marked as "suspect".

**T-ALL subtype annotation**. T-ALL subtype annotations were downloaded from the St. Jude's PeCan Portal (https://pecan.stjude.cloud/proteinpaint/study/target-tall) which displays mutation data of T-ALL tumors in a large genomics study[31]. Of all 266 T-ALL samples used in our data set, we could assign 263 tumors a subtype category (Ordered by maturation stage: LMO2/LYL1: 18, HOXA: 33, TLX3: 46, TLX1: 26, NKX2-1: 14, LMO1/2: 10, TAL1: 86, TAL2: 8, Unknown: 22). We removed the three unassigned tumors, and those from the 'unknown' category before generating Fig. 5.

**Candidate Reporting web application**. We developed an R (v3.5.2) web application with Shiny (v1.5.0) to provide visualizations and additional up-to-date variant annotation of genetic interaction candidates. We first ran the Ensembl Variant Effect Predictor (VEP)[47] GRCh37 (v101) on the original set of mutations to predict the genes, transcripts, protein sequences, and their consequences. We restricted results by selecting VEP fields indicating the impact of the variant (e.g., Consequence and Condel) and by choosing one consequence per variant, which translates to the top-ranking transcript. In addition, fields indicating the cancer type and the source data set were added to the VEP output and the final table was translated to a Mutation Annotation Format (MAF) object. Maftools[48] (v1.8.10) was used to create lollipop plots, oncoplots, and data summary plots from subsets of the MAF object by filtering on the cancer type and data set. One-letter code amino acid changes for the lollipop plots were derived from the protein sequence in the HGVS recommended format (HGVSp field of VEP). The Candidate Reporting web application can be found on https://gi-analysis.kemmerenlab.eu/.

After running VEP annotation for the reporting tool, we noticed minor changes in gene annotation in candidate gene pairs compared to the original mutation files. Six patients (three in T-ALL, one in AML, and one in B-ALL) had mutations previously annotated in CNKSR3, while these were actually mutations in the gene MAGI1. This affected all (three) T-ALL patients that were involved in the mutually exclusive CNKSR3-NOTCH1, and we, therefore, changed this candidate to MAGI1-NOTCH1. Furthermore, we found three patients (one in T-ALL and two in NBL) with mutations previously annotated in PTEN where these should be mutations in TEP1. PTEN is part of several T-ALL candidate genetic interactions. We, therefore, repeated the WeSME test in T-ALL ($N = 20$ runs), changing this one patient mutation from PTEN to TEP1. Some interactions (NOTCH1-USP7, PHGS-PIK3R1, PHF6-PTEN, and USP7-WT1) scored lower with significance in 17 out of

20 runs (compared to significance in 9 or 10 out of 10 runs previously). As these are still high confidence candidates, we kept them in our final result set.

## Data availability

The somatic mutation data that support our findings are available via the National Cancer Institute TARGET Data Matrix (https://ocg.cancer.gov/programs/target/data-matrix) and the R2 DKFZ Pediatric Cancer Data Portal (https://hgserver1.amc.nl/cgi-bin/r2/main.cgi?option=about_dscope). We further refer to the original studies[6,7] for more information on the data availability of the raw data files. Source data underlying main figures can be found in Supplementary Data 1–2. Processed data files that served as input for our genetic interaction pipeline (gene–sample mutation matrices) are available upon request.

## Code availability

Figure 3 was created with the network visualization software Cytoscape (version 3.7.1). Cancer type-specific candidates were validated with the R version of the DISCOVER method (version 0.9.2) (https://ccb.nki.nl/software/discover/). R code to run the Permutation test and the modified WeSME Python scripts are available on the Github repository https://github.com/princessmaximacenter/GI_InteractionTests. The code for the R Shiny Candidate Reporting tool is available on https://github.com/princessmaximacenter/GI_ReportingTool.

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

## Acknowledgements

We would like to thank Jarno Drost, Kim Verhagen, Jules Meijerink, Esther Hulleman, Judith Boer, Frank van Leeuwen and members of the lab for fruitful discussions and feedback. Financial support has been provided by the Dutch Cancer Society (grant no. 10354), as well as by the Dutch Organization for Scientific Research (NWO; grant no. 864.11.010) and KiKa.

## Author contributions

P.K., F.H. and J.D. designed the study. J.D. and S.A. designed and implemented the statistical pipeline and analyzed and interpreted test results. D.K. designed and implemented the candidate reporting tool. S.M., N.J., J.Z. and S.P. provided the data set and assisted in interpretation of the data. J.D., S.A., P.K. and D.K. prepared the paper. J.Z., S.P. and F.H. edited the paper.

## Competing interests

The authors declare no competing interests.
