## [Transparent Peer Review File · Communications Biology]

Reviewers' comments:

Reviewer #1 (Remarks to the Author):

In this study, the authors analyzed mutation data on paediatric cancers from two databases, DKFZ and TARGET, including 2500 tumors and 23 tumor types. The authors reported 15 co-occurring, 27 mutually exclusive gene pairs that suggest functional interactions between genes in each pair. They explained the mutually exclusivity by tumor subtypes, pathway epistasis, and synthetic lethality.

This reviewer has the following concerns on the study.

No results are replicated among different tumor types or between the two datasets. It is understandable that different cancers may have different mutation profiles. On the other hand, genetic interactions can still have certain commonalities and I suspect that if we have enough samples, certain interactions might be true across tumor types. The authors explained the lack of replication from DKFZ dataset on findings from TARGET by smaller sample size. It is acknowledged that sample size is smaller in DKFZ, but certain similar signals are still expected. If you randomly sample the TARGET dataset with a third of the total samples, how often the same pairs will be observed? Is the lack of replication a suggestion that the interactions are disproportionately affected by small number of samples? If the results can not be reproduced by additional samples, does that mean that the possibility of artifacts can not be ruled out?

The explanations on the mechanisms of the interaction pairs read more like a discussion rather than real analysis results. They are all plausible but treating them as they are and coming up with a percentage of underlying mechanism estimation is not very convincing. These interpretations are also not mutually exclusive. Pathway epistasis is more like functional redundancy.

The two analysis methods are very similar to each other and in a way they do not serve the benefit of complementing each other in terms of statistical rigor.

I don't question the probability that some of the gene pairs are truly interaction partners functionally. However, I don't get a good sense of adequate study power in the face of enormous heterogeneity of the cancer even when they belong to the same type and I am not sure to what extent that the findings can be generalized, and whether the FDRs and P values really serve as evaluation parameters of the findings. It is suspected that some of the mutually exclusivity might just be a reflection of lack of detection power.

Reviewer #2 (Remarks to the Author):

The authors applied a statistical pipeline to detect genetic interactions in a combined dataset comprising over 2,500 tumors from 23 cancer types. They found a genetic interaction map of childhood cancers comprising 15 co-occurring and 27 mutually exclusive candidates. They further elucidated biological mechanisms underlying most candidates are either tumor subtype, pathway epistasis or cooperation while synthetic lethality plays a much smaller role. They concluded that other explanations beyond synthetic lethality should be considered when interpreting results of genetic interaction tests. The work has some merits in the related field. The webtool seems useful. I have a few comments.

1. The title seems too big to reflect the actual study. It should emphasize the analysis part, something like statistical analysis of genetic maps reveal....
2. The Results part should be shortened. Many are redundant from the Methods part. For example, the 1st and 2nd sections are mostly used for describing the data sets and statistical methods.
3. The meaning of biological mechanisms here is very ambiguous. I am not quite sure why the authors

refer the tumor subtype, pathway epistasis, cooperation and synthetic lethality as some sorts of biological mechanisms.

4. Most of these pairs are between driver mutation genes. Have the authors statistically examined the pairs of driver mutation genes and computationally identified passenger mutation genes by other programs? In this case, they may find some novel interaction.

Response to reviewers manuscript COMMSBIO-20-3512-T

Reviewer #1 (Remarks to the Author):

In this study, the authors analyzed mutation data on paediatric cancers from two databases, DKFZ and TARGET, including 2500 tumors and 23 tumor types. The authors reported 15 co-occurring, 27 mutually exclusive gene pairs that suggest functional interactions between genes in each pair. They explained the mutual exclusivity by tumor subtypes, pathway epistasis, and synthetic lethality.

This reviewer has the following concerns on the study.

1. No results are replicated among different tumor types or between the two datasets. It is understandable that different cancers may have different mutation profiles. On the other hand, genetic interactions can still have certain commonalities and I suspect that if we have enough samples, certain interactions might be true across tumor types. The authors explained the lack of replication from DKFZ dataset on findings from TARGET by smaller sample size. It is acknowledged that sample size is smaller in DKFZ, but certain similar signals are still expected. If you randomly sample the TARGET dataset with a third of the total samples, how often the same pairs will be observed? Is the lack of replication a suggestion that the interactions are disproportionately affected by small number of samples? If the results cannot be reproduced by additional samples, does that mean that the possibility of artifacts cannot be ruled out?

The reviewer suggests to randomly subsample the TARGET dataset to see how often the same pairs are observed. This is a great suggestion and exactly what we did. The results are described in paragraph "Sample size explains lack of overlap in candidates for cancer types present in both data sets". Our main finding: performing the genetic interaction test on a down-sampled TARGET data set results almost always in zero high-confidence candidates (replicated at least 9 out of ten times). We conclude that with smaller sample size you lose power to detect the candidates found in the original test. As stated in the manuscript we think that this also explains to a large degree the lack of overlap between the two datasets as the number of samples per cancer type is lower for the DKFZ dataset. Of course, we cannot rule out that some of the candidates are false positives. Indeed, during our investigation of possible explanations for our findings, we consider several candidates to be most likely false positives as they do not show strong evidence for any biological mechanism that would explain the mutual exclusivity or co-occurrence. They are marked as such in Table 2.

2. The explanations on the mechanisms of the interaction pairs read more like a discussion rather than real analysis results. They are all plausible but treating them as they are and coming up with a percentage of underlying mechanism estimation is not very convincing. These interpretations are also not mutually exclusive. Pathway epistasis is more like functional redundancy.

We agree that the explanation of possible mechanisms underlying the candidate pairs have a more interpretative character than a mere presentation of the results of our analyses. At the same time, we do want to stress that the listing of possible mechanisms is the result of a thorough literature review, taking up a large part of the whole project. To make this more explicit, we have now added a Supplementary Table S4 that provides for each candidate gene pair the relevant literature used and more detailed explanation how we derived the proposed mechanism and also make this more explicit in the manuscript.

The reviewer states that the interpretations are not mutually exclusive. We agree that for several gene pairs more than one mechanism could explain the results. In Table 2, we have selected the most

likely explanation, but we do elaborate in the new supplemental table S4 what alternative explanations exist for some of the candidate gene pairs.

However, we do think that one mechanism is usually the driving mechanism/explanation for the found mutually exclusivity or co-occurrence of a given candidate gene pair. For example, if a mutated gene pair has a synthetically lethal relationship, it is unlikely that the same mutations are functionally redundant (which is by the way indeed what we mean with pathway epistasis).

3. The two analysis methods are very similar to each other and in a way they do not serve the benefit of complementing each other in terms of statistical rigor.

We completely agree with the reviewer on this point. In our manuscript, we also do not claim that the two methods complement each other, but rather show that the WeSME test is a good alternative for the permutation test as it produces highly similar results in less computational time.

We have now added the results of an additional complimentary test, DISCOVER (Canisius et al. 2016), which also uses a binary gene-sample matrix as input but implements a different statistical strategy to find mutually exclusive and co-occurring gene pairs. We show that applying DISCOVER on the TARGET and DKFZ data sets results in a very similar candidate set. We have added the DISCOVER results to Supplementary Table S2 and included a section in the discussion to reflect on this. Note that DISCOVER does not have functionality to run a PAN cancer test, so we could not validate our PAN cancer results with this method.

4. I don't question the probability that some of the gene pairs are truly interaction partners functionally. However, I don't get a good sense of adequate study power in the face of enormous heterogeneity of the cancer even when they belong to the same type and I am not sure to what extent that the findings can be generalized, and whether the FDRs and P values really serve as evaluation parameters of the findings. It is suspected that some of the mutually exclusivity might just a reflection of lack of detection power.

We disagree with the reviewer that mutually exclusivity might reflect a lack of detection power. In the Permutation and WesME test, a mutated gene pair is considered to be significantly mutually exclusive when the co-occurrence count is significantly lower than expected given the mutation frequency of each gene individually. So, a significant mutually exclusive candidate occurs when both genes have a high enough mutation rate individually to have an expected co-occurrence count larger than zero. In general, the test would show the opposite: due to a lack of power it might miss potential genetic interactions. Through down-sampling of the TARGET data set, the tests indeed show less power with smaller sample sizes.

We use a randomization approach to build a null distribution that actually reflects the heterogeneity (variability in gene- and tumor-level mutation rates) within a cancer type. The empirical P-values and FDRs based on this null distribution are in our view the correct measures to infer significance. Many other studies perform co-occurrence and mutual exclusivity analyses based on more general methods, such as a fisher exact test and standard FDR calculations. These approaches do not take the underlying structure of the data set into account which can lead to more false positives (fisher exact test) or more false negatives (standard FDR calculations). We have added some additional sentences in the Methods section to make this clearer.

Reviewer #2 (Remarks to the Author):

The authors applied a statistical pipeline to detect genetic interactions in a combined dataset comprising over 2,500 tumors from 23 cancer types. They found a genetic interaction map of childhood cancers comprises 15 co-occurring and 27 mutually exclusive candidates. They further elucidated biological mechanisms underlying most candidates are either tumor subtype, pathway epistasis or cooperation while synthetic lethality plays a much smaller role. They concluded that other explanations beyond synthetic lethality should be considered when interpreting results of genetic interaction tests. The work has some merits in the related field. The webtool seems useful. I have a few comments.

1. The title seems too big to reflect the actual study. It should emphasize the analysis part, something like statistical analysis of genetic maps reveal....

We consider our study to be unique on two aspects: First, as far as we know our study is the first to present a comprehensive pan-cancer genetic interaction analysis in childhood cancer. Second, in contrast with other 'genetic interactions in cancer' studies that usually end up with presenting a list of possible candidate gene pairs, we thoroughly investigate the underlying mechanisms that can explain the found co-occurrence and mutual exclusivity and show that synthetic lethality plays a much smaller part than previously assumed/hoped. With our title we aim to capture these two results. Based on the suggestion of the reviewer, the title now states "A systematic analysis of genetic interactions reveals the underlying biology in childhood cancer" and better emphasizes the analysis part of the study.

2. The Results part should be shorten. Many are redundant from the Methods part. For example, the 1st and 2nd sections are mostly used for describing the data sets and statistical methods.

We aim to have a Results section that can be read independently from the methods sections. Therefore, we think it is good practice to briefly describe the data and methods at the start of the results section. However, we agree with the reviewer that the description of the data set and methods could be shortened. In the revised manuscript, we have reduced both sections extensively.

3. The meaning of biological mechanisms here is very ambiguous. I am not quite sure why the authors refer the tumor subtype, pathway epistasis, cooperation and synthetic lethality as some sorts of biological mechanisms.

We used the term 'biological mechanisms' as opposed to technical artifacts. We have now replaced 'biological mechanism' with the term 'biological explanation'.

4. Most of these pairs are between driver mutation genes. Have the authors statistically examined the pairs of driver mutation genes and computationally identified passenger mutation genes by other programs? In this case, they may find some novel interaction.

This is an excellent suggestion by the reviewer and something we have incorporated in our analyses and made more explicit in our manuscript. In both the permutation test and WeSME test we have included all genes that were mutated in more than one patient in a cancer type, so more than only driver genes. However, we show that the resulting candidate gene pairs in almost all cases consist of SMGs (significantly mutated genes, most likely driver genes). This makes sense, since only gene pairs with sufficient mutation rates can be detected in our statistical pipeline. However, we did find candidates involving non-SMGs: for instance, the CTNNB1 (SMG) - EFCAB6 (non-SMG) gene pair in Wilms Tumor and NOTCH1 (SMG) - MAGI1 (non-SMG) in T-ALL. Note that these SMGs were annotated as such when the DKFZ and TARGET data sets were created, using common tools such as MuSic.

Reviewers' comments:

Reviewer #1 (Remarks to the Author):

The authors have tried their best to address my comments and I appreciate the effort. Clearly there is value in the study, while the limitations are also obvious. I think it is the editor's call to compare the manuscript with what is usually published in this journal to evaluate whether to accept the manuscript. Clearly it will be good to acknowledge the limitations.

Reviewer #2 (Remarks to the Author):

The authors have now addressed my comments.

Response to reviewers manuscript COMMSBIO-20-3512-A

Reviewer #1 (Remarks to the Author):

The authors have tried their best to address my comments and I appreciate the effort. Clearly there is value in the study, while the limitations are also obvious. I think it is the editor's call to compare the manuscript with what is usually published in this journal to evaluate whether to accept the manuscript. Clearly it will be good to acknowledge the limitations.

We appreciate the comment from reviewer #1 and are in particular grateful that the reviewer also sees the value of the study. Based on this comment, together with the additional comments from the editor, we have added additional sentences throughout the discussion that clearly state the limitations of the study with regard to cohort size, cancer heterogeneity and lack of functional validation.

Reviewer #2 (Remarks to the Author):

The authors have now addressed my comments.